# Subquadratic High-Dimensional Hierarchical Clustering

**Amir Abboud**
IBM Research
amir.abboud@gmail.com

**Vincent Cohen-Addad**
CNRS & Sorbonne Université
vcohenad@gmail.com

**Hussein Houdrouge**
École Polytechnique
hussein.houdrouge@polytechnique.edu

## Abstract

We consider the widely-used average-linkage, single-linkage, and Ward's methods for computing hierarchical clusterings of high-dimensional Euclidean inputs. It is easy to show that there is no efficient implementation of these algorithms in high dimensional Euclidean space since it implicitly requires to solve the closest pair problem, a notoriously difficult problem.

However, how fast can these algorithms be implemented if we allow approximation? More precisely: these algorithms successively merge the clusters that are at closest average (for average-linkage), minimum distance (for single-linkage), or inducing the least sum-of-square error (for Ward's). We ask whether one could obtain a significant running-time improvement if the algorithm can merge $\gamma$-approximate closest clusters (namely, clusters that are at distance (average, minimum, or sum-of-square error) at most $\gamma$ times the distance of the closest clusters).

We show that one can indeed take advantage of the relaxation and compute the approximate hierarchical clustering tree using $\widetilde{O}(n)$ $\gamma$-approximate nearest neighbor queries. This leads to an algorithm running in time $\widetilde{O}(nd) + n^{1+O(1/\gamma)}$ for $d$-dimensional Euclidean space. We then provide experiments showing that these algorithms perform as well as the non-approximate version for classic classification tasks while achieving a significant speed-up.

## 1 Introduction

Hierarchical Clustering (HC) is a ubiquitous task in data science. Given a data set of $n$ points with some similarity or distance function over them, the goal is to group similar points together into clusters, and then recursively group similar clusters into larger clusters. The clusters produced throughout the procedure can be thought of as a hierarchy or a tree with the data points at the leaves and each internal node corresponds to a cluster containing the points in its subtree. This tree is often referred to as a "dendrogram" and is an important illustrative aid in many settings. By inspecting the tree at different levels we get partitions of the data points to varying degrees of granularity. Famous applications are in image and text classification [39], community detection [28], finance [40], and in biology [8, 19].

Perhaps the most popular procedures for HC are Single-Linkage, Average-Linkage, and Ward's method. These are so-called *agglomerative* HC algorithms (as opposed to *divisive*) since they proceed in a bottom-up fashion: In the beginning, each data point is in its own cluster, and then the

most similar clusters are iteratively merged - creating a larger cluster that contains the union of the points from the two smaller clusters - until all points are in the same, final cluster.

The difference between the different procedures is in their notion of similarity between clusters, which determines the choice of clusters to be merged. In Single-Linkage the distance (or *dissimilarity*) is defined as the minimum distance between any two points, one from each cluster. In Average-Linkage we take the average instead of the minimum, and in Ward's method we take the error sum-of-squares (ESS). It is widely accepted that Single-Linkage enjoys implementations that are somewhat simpler and faster than Average-Linkage and Ward's, but the results of the latter two are often more meaningful. This is because its notion of distance is too sensitive and a meaningless "chain" in the data can sabotage the resulting clustering. Extensive discussions of these procedures can be found in many books (e.g. [21, 28, 37, 1]), surveys (e.g. [31, 32, 9]), and experimental studies (e.g. [34]).

All of these procedures can be performed in nearly quadratic time, and the main question studied by this paper is whether we can reduce the time complexity to subquadratic. The standard quadratic algorithm for Single-Linkage is quite simple and can be described as follows. After computing the $n \times n$ distance matrix of the points, we find a minimum spanning tree (MST). This first stage takes $O(n^2 d)$ time if the points are in $d$-dimensional Euclidean space. In the second stage we perform merging iterations, in which the clusters correspond to connected subgraphs of the MST (initially, each point is its own subgraph). We merge the two subgraph whose in-between edge in the MST is the smallest. By the properties of MST, the edge between two subgraphs (clusters) is exactly the minimum distance between them. This second stage can be done with $O(n)$ insertions, deletions, and minimum queries to a data structure, which can be done in near-linear time. The algorithms for Average-Linkage and Ward's are more complicated since the MST edges between two clusters can be arbitrarily smaller than the average distance or the ESS between them, and we must consider all pairwise distances in clusters that quickly become very large. Nonetheless, an $O(n^2 \log n)$ algorithm (following a first stage of computing the distance matrix) has been known for many decades [31].

Can we possibly beat quadratic time? It is often claimed (informally) that $\Omega(n^2)$ is a lower bound because of the fist stage: it seems necessary to compute the distance matrix of the points whose size is already quadratic. More formally, we observe that these procedures are at least as hard as finding the *closest pair* among the set of points, since the very first pair to be merged *is* the closest pair. And indeed, under plausible complexity theoretic assumptions[1], there is an almost-quadratic $n^{2-o(1)}$ lower bound for the closest pair problem in Euclidean space with dimension $d = \omega(\log n)$ [2, 26]. This gives a quadratic conditional lower bound for all three Single-Linkage, Average-Linkage, and Ward's method.

Achieving subquadratic runtime has been of interest for many decades (as can be deduced from the survey of Murtagh [31]) and it is increasingly desirable in the era of big data. (See also the recent work on quadratic vs. subquadratic complexity of Empirical Risk Minimization problems [5].)

In this work, we focus on worst-case guarantees while allowing for small approximation in the answers: how fast can we perform these procedures if each iteration is allowed to pick an *approximately best* pair to merge? More precisely, when merging two clusters the algorithm is allowed to do the following. If the best pair of (available) clusters has (minimum, average, or ESS) distance $d$ then the algorithm can choose any pair of clusters whose distance is between $d$ and $\gamma \cdot d$, where $\gamma \geqslant 1$ is a small constant.

When approximations are allowed the time complexity of closest pair drops, and so does the conditional lower bound. Even in high dimensions, Locality Sensitive Hashing techniques can find the $\gamma$-approximate nearest neighbors (ANN) in $L_1$-distance with $n^{O(1/\gamma)}$ time per query [3, 4]. This gives a subqadratic $n^{1+O(1/\gamma)}$ algorithm for closest pair[2], but can we achieve the same speed-up for $\gamma$-approximate Average-Linkage? Namely, can we do Average-Linkage as fast as performing $\widetilde{O}(n)$ (approximate) nearest-neighbor queries?

For the simpler $\gamma$-approximate Single-Linkage it is rather easy to see that the answer is yes. This essentially follows from the classical Nearest Neighbor Chain algorithm for HC [31]. Here is a

simple way to see why subquadratic is possible in this case: The idea is to replace the expensive first stage of the Single-Linkage algorithm (described above) with an approximate MST computation which can be done in subquadratic time [7, 22] using ANN queries. Then we continue to perform the second stage of the algorithm with this tree.

Still it is of great interest to speed up the Average-Linkage and Ward's algorithms since they typically give more meaningful results. This is much harder and before this work, no subquadratic time algorithm for Average-Linkage or Ward's with provable guarantees were known. Various algorithms and heuristics have been proposed, see e.g. [38, 20, 27, 33, 25, 41], that beat quadratic time by either making assumptions on the data or by changing the merging criteria altogether. Intuitively, while in Single-Linkage only $O(n)$ distances are sufficient for the entire computation (the distances in the MST), it is far from clear why this would be true for Average-Linkage and Ward's.

## 1.1 Our Contribution

Our main results are a $\gamma$-approximate Ward's algorithm, and a $\gamma$-approximate Average-Linkage algorithm that run in subquadratic $\tilde{O}(n^{1+O(1/\gamma)} + nd)$ time, for any $\gamma > 1$, when the points are in $d$-dimensional Euclidean space. Moreover, our algorithms are reductions to $\tilde{O}(n)$ approximate nearest neighbor queries in dimension $\tilde{O}(d)$ with $L_2$ distance squared (for Ward's) or $L_1$ distance (for Average-Linkage). Thus, further improvements in ANN algorithms imply faster approximate HC, and more importantly, one can use the optimized ANN libraries to speed up our algorithm in a black-box way. In fact, this is what we do to produce our experimental results. Our theorems are as follows.

**Theorem 1.1.** *Given a set of $n$ points in $\mathbb{R}^d$ and a $\sqrt{\gamma}$-Approximate Nearest Neighbor data structure which supports insertion, deletion and query time in time $T$, there exists a $\gamma(1 + \varepsilon)$-approximation of Ward's Method running in time $O(n \cdot T \cdot \varepsilon^{-2} \log(\Delta n) \log n)$, where $\Delta$ is the aspect ratio of the point set.*

**Theorem 1.2.** *Given a set of $n$ points in $\mathbb{R}^d$ and a data structure for $\gamma$-Approximate Nearest Neighbor under the $L_1$-norm which supports insertion, deletion and query time in time $T$, there exists a $\gamma(1 + \varepsilon)$-approximation of Average Linkage running in time $n \cdot T \cdot \varepsilon^{-2} \log^{O(1)}(\Delta n)$, where $\Delta$ is the aspect ratio of the point set.*

Our algorithm for approximating Ward's method is very simple: We follow Ward's algorithm and iteratively merge clusters. To do so efficiently, we maintain the list of centroids of the current clusters and perform approximate nearest neighbor queries on the centroids to find the closest clusters. Of course, this may not be enough since some clusters may be of very large size compared to others and this has to be taken into account in order to obtain a $\gamma$-approximation. We thus partition the centroids of the clusters into buckets that represents the approximate sizes of the corresponding clusters and have approximate nearest neighbor data structure for each bucket. Then, given a cluster $C$, we identify its closest neighbor (in terms of Ward's objective) by performing an approximate nearest neighbor query on the centroid of $C$ for each bucket and return the best one.

Our algorithm for Average-Linkage is slightly more involved. Our algorithm adapts the standard Average-Linkage algorithm, with a careful sampling scheme that picks out representatives for each large cluster, and a strategic policy for when to recompute nearest neighbor information. The other sections of this paper are dedicated to explaining the algorithm. Implementation-wise it is on the same order of complexity as the standard Average-Linkage algorithm (assuming a nearest neighbor data structure is used as a black-box), while efficiency-wise it is significantly better as it goes below quadratic time. The gains increase as we increase the tolerance for error, in a controlled way.

We focus our empirical analysis on Ward's method. We show that even for a set of parameters inducing very loose approximation guarantees, the hierarchical clustering tree output by our algorithm is as good as the hierarchical clustering tree produced by Ward's method in terms of classification for most of several classic datasets. On the other hand, we show that even for moderately large datasets, e.g.: sets of 20-dimensional points of size 20000, our algorithm offers a speed-up of 2.5 over the popular implementation of Ward's method of sci-kit learn.

## 1.2 Related Works

A related but orthogonal approach to ours was taken by a recent paper [14]. The authors design an agglomerative hierarchical clustering algorithm, also using LSH techniques, that at each step, with constant probability, performs the merge that average linkage would have done. However, with constant probability, the merge done by their algorithm is arbitrary, and there is no guarantee on the quality of the merge (in terms of average distance between the clusters merged compared to the closest pair of clusters). We believe that our approach may be more robust since we have a guarantee on the quality of every merge, which is the crux of our algorithms. Moreover, they only consider Average-Linkage but not Ward's method.

Strengthening the theoretical foundations for HC has always been of interest. Recently, an influential paper of Dasgupta [17] pointed to the lack of a well-defined objective function that HC algorithms try to optimize and proposed one such function. Follow up works showed that Average-Linkage achieves a constant factor approximation to (the dual of) this function [16, 29], and also proposed new polynomial time HC algorithms for both worst-case and beyond-worst-case scenarios that can achieve better approximation factors [35, 10, 15, 11, 12]. Other theoretical works prove that Average-Linkage can reproduce a "correct" clustering, under some stability assumptions on the data [6]. Our work takes a different approach. Rather than studying the reasons for the widespread empirical findings of the utility of HC algorithms (and mainly Average-Linkage and Ward's), we take it as a given and ask: how fast can we produce results that are as close as possible to the output of Average-Linkage and Ward's. In some sense, the objective function we try to optimize is closeness to whatever Average-Linkage or Ward's produce.

## 1.3 On our Notion of Approximation

The approximate Average-Linkage notion that we define ($\gamma$-AL) guarantees that at every step, the merged pair is $\gamma$-close to the best one. But can we prove any guarantees on the quality of the final tree? Will it be "close" to the output of (exact) AL? (The same applies for Ward's, but let us focus on AL in this subsection.)

One approach is to look at certain objective functions that measure the quality of a hierarchical clustering tree, such as the ones mentioned above ([16, 29] and by [17] for similarity graphs), and compare the guarantees of AL and our $\gamma$-AL w.r.t. these objective functions. It is likely that one can prove that $\gamma$-AL is guaranteed to give a solution that is no worse than an $O(\gamma)$ factor from the guarantees of (exact) AL w.r.t. to these objective functions. However, such a theorem may not have much value because (as shown by Charikar et al. [11]) the guarantees of AL are no better than those of a random recursive partitioning of the dataset. Therefore, such a theorem will only prove that $\gamma$-AL is not-much-worse than random, which dramatically understates the quality of $\gamma$-AL. In fact, in our experiments with a standard classification task, $\gamma$-Ward's is very close to Ward's and is *much* better than random (random has a $1/k$ success rate, which is $0.1$ or less in case of digits, while ours achieves $0.5 - 0.8$).

Another approach would be to prove theorems pertaining to an objective function for HC that offers the guarantee that given two trees, if their costs are close then the structures of their HCs are similar. Unfortunately, we are not aware of any such objective functions (this is also the case for flat clusterings such as k-median, k-means, etc.). In particular, with the functions of [16, 29] the trees output by AL and by a random recursive partitioning have the same cost, while their structure may be very different.

Besides the empirical evidence, let us mention two more points in support of our algorithms. First, our algorithms are essentially reductions to Approximate Nearest Neighbor (ANN) queries, and ANN queries (using LSH for example) perform very well in practice. In fact, on real world inputs, the algorithm often identifies the *exact* nearest neighbor and then performs the same merge as in AL. Second, we can provide a theoretical analysis of the following form in support of $\gamma$-AL. It is known that if the input data is an ultrametric, then AL (and also Single-Linkage or Complete-Linkage) does recover the underlying ultrametric tree (see e.g. [16]) . Now, assume that the ultrametric is *clear* in the sense that if $d(a, b) > d(a, c)$ then $d(a, b) > \gamma d(a, c)$ for some constant $\gamma$. In this case, our algorithm will provably recover the ultrametric in $n^{1+O(1/\gamma)}$ time, whereas AL would need $\Omega(n^2)$ time. Notably, in this setting, obtaining an $O(1)$-approximation w.r.t. the objective functions of [16, 29] does not mean that the solution is close to the ultrametric tree.

## 2 A $\gamma$-Approximation of Ward's Method

### 2.1 Preliminaries

Let $P \subset R^d$ be a set of $n$ points. Up to rescaling distances we may assume that the minimum distance between any pair of points is 1. Let $\Delta$ denote the aspect ratio of $P$, namely $\Delta = \max_{u,v \in P} \text{dist}(u,v)$. Let $\gamma > 1$ be a fixed parameter. Our goal is to build a $\gamma$-approximation of Ward's hierarchical clustering.

Let $C$ be a cluster, then define the *error sum-of-square* as

$$ESS(C) = \sum_{x \in C} (x - \mu(C))^T (x - \mu(C))$$

where $\mu(C) = \frac{1}{|C|} \sum_{x \in C} x$. We let the *error sum-of-square of a clustering* $\mathcal{C} = \{C_1, \ldots, C_\ell\}$ be

$$ESS(\mathcal{C}) = \sum_{C \in \mathcal{C}} ESS(C).$$

Thus, Ward's algorithm constructs a hierarchy of clusters where each level represents a clustering of the points and where clusters at a given level $\ell$ are subsets of clusters of level $\ell + 1$. Ward's algorithm builds this hierarchy in a bottom-up fashion, starting from $n$ clusters (each point is itself a cluster). Then, given the clustering of a given level $\ell$, Ward's algorithm obtains the clustering of the next level by merging the two clusters that yield the clustering of minimal ESS. More formally, consider a clustering $\mathcal{C} = \{C_1, \ldots, C_\ell\}$. To find the clustering of minimum ESS obtained by merging a pair of clusters of $\mathcal{C}$, it is enough to minimize the increase in the ESS induced by the merge. Therefore, we want to identify the clusters $C_i, C_j$ that minimize the following quantity.

$$\Delta ESS(C_i, C_j) = \frac{|C_i||C_j|}{|C_i| + |C_j|} ||\mu(C_i) - \mu(C_j)||_2^2. \tag{1}$$

We will also make use of the following fact.

**Fact 1.** *Given two set of points $A, B$ with corresponding centroids $\mu(A), \mu(B)$ respectively, we have that the centroid of $A \cup B$ is on the line joining $\mu(A)$ to $\mu(B)$, at distance $\frac{|B|}{|A \cup B|} ||\mu(A) - \mu(B)||_2^2$ from $\mu(A)$.*

Let $\gamma > 0$ be a parameter, $P$ a set of points in $\mathbb{R}^d$. Let $\mathcal{D}$ be a data structure that for any set $P$ of n points in $\mathbb{R}^d$ where $d = O(\log n)$, supports the following operations. Insertion of a point in $P$ in time $O(n^{f(\gamma)})$, for some function $f$. Deletion of a point in $P$ in time $O(n^{f(\gamma)})$; Given a point $p \in P$, outputs a point inserted to the data structure at $L_2 - distance$ at most $\gamma$ times the distance from $p$ to the closest point inserted to the data structure, in time $O(n^{f(\gamma)})$.

There are data structures based on locality sensitive hashing for $f(\gamma) = 1 + O(1/\gamma^2)$, see for example [4].

#### 2.1.1 Finding The Nearest Neighbour Cluster

Our algorithm relies on a Nearest Neighbour Data Structure for clusters, where the distance between two clusters $A, B$ is given by $ESS(A \cup B) - ESS(A) - ESS(B)$. Given a parameter $\varepsilon > 0$, our Nearest Neighbour Data Structure $\mathcal{D}(\gamma, \varepsilon)$ for clusters consists of $O(\varepsilon^{-1} \log n)$ Nearest Neighbour Data Structures for points with error parameter $\sqrt{\gamma}$ defined as follows. There is a data structure $\mathcal{D}^\ell$ for each $\ell \in \{(1 + \epsilon)^i \mid i \in [1, \ldots, \log_{1+\epsilon} n]\}$. The data structure works as follows.
**Insertion($C$):** Inserting a cluster of a set $C$ of points is done by inserting $\mu(C)$ in the $\mathcal{D}^i$ such that $(1 + \varepsilon)^{i-1} \leqslant |C| < (1 + \varepsilon)^i$.
**Query($C$):** For each $i \in \{(1 + \epsilon)^i \mid i \in [1, \ldots, \log_{1+\epsilon} n]\}$ perform a nearest neighbor data query for $\mu(C)$ in $D^i$, let $NN_i(C)$ be the result. Output $NN_i(C)$ that minimizes $\Delta ESS_{C, NN_i(C)}$.

The proof of the following lemma is in the appendix.

**Lemma 2.1.** *For any $\varepsilon > 0$, the above nearest neighbour data structure for clusters with parameters $\gamma, \varepsilon, \mathcal{D}(\gamma, \varepsilon)$ has the following properties:*

- *The insertion time is $O(n^{f(\sqrt{\gamma})}\varepsilon^{-1}\log n)$;*

- *On Query(C), it returns a clusters $C'$ such that $ESS(C \cup C') - ESS(C) - ESS(C') \leqslant$ $(1+\varepsilon)\gamma\min_{B\in\mathcal{D}(\varepsilon,\gamma)}(ESS(C \cup B) - ESS(C) - ESS(B))$.*

- *The query time is $O(n^{f(\sqrt{\gamma})}\varepsilon^{-1}\log(n\Delta))$.*

### 2.1.2 The Main Algorithm

We define the *value of merging two clusters $A,B$* as $ESS(A \cup B) - ESS(A) - ESS(B)$. Our algorithm starts by considering each point as its own cluster, together with the Nearest Neighbour Cluster Data Structure described above. Then, the algorithm creates a logarithmic number of *rounded merge values* that partition the range of possible merge values. Let $\mathcal{I}$ be the sequence of all possible merge values in increasing order.

Given a set of $n$ points with minimum pairwise distance 1 and maximum pairwise distance $\Delta$, we have that the total number of merge value $\beta$ is $O(\log(n\Delta))$.

The algorithm maintains a clustering and at each step decides which two clusters of the current clustering should be merged. The clusters of the current clustering are called *unmerged clusters*. The algorithm iterates over all merge values in an increasing order while maintaining the following invariant:

**Invariant 2.2.** *When the algorithm reaches merge value $\delta$, for any pair of unmerged cluster $C, C'$ we have $ESS(C \cup C') - ESS(C) - ESS(C') \geqslant \delta/\gamma$.*

We now give a complete description of our algorithm.

1. Let $\mathcal{L}$ be the list of unmerged clusters, initially it contains all the points.
2. For each $\nu \in \mathcal{I}$:
   (a) ToMerge $\leftarrow \mathcal{L}$
   (b) While ToMerge is not empty:
       i. Pick a cluster $C$ from ToMerge, and remove it from ToMerge.
       ii. $NN(C) \leftarrow$ Approximate Nearest Neighbour Cluster of $C$.
       iii. If $ESS(C \cup NN(C)) - ESS(C) - ESS(NN(C)) \leqslant \nu$:
           A. Merge $C$ and $NN(C)$. Let $C'$ be the resulting cluster.
           B. Remove $NN(C)$ from ToMerge and add $C'$ to ToMerge; $\mu(C')$ follows immediately from $\mu(C), \mu(NN(C)), |C|$ and $|NN(C)|$ (see Fact 1)
           C. Remove $C, NN(C)$ from $\mathcal{L}$ and add $C'$ to $\mathcal{L}$

The running time analysis and proof of correctness of the algorithm are deferred to the appendix.

## 3 A $\gamma$-Approximation of Average-Linkage

### 3.1 Preliminaries

For two sets of points $A, B$, we let $\text{avg}(A, B) = \frac{1}{|A||B|}\sum_{a\in A}\sum_{b\in B} d(a,b)$. The following simple lemma is proved in the appendix.

**Lemma 3.1.** *Consider three sets of points $A, B, C$. We have that $\text{avg}(A,C) = \text{avg}(C,A) \leqslant \text{avg}(A,B) + \text{avg}(B,C)$*

### 3.2 Overview and Main Data Structures

Our goal is to design a $\gamma$-approximate Average-Linkage algorithm. The input is a set $P$ of $n$ points in a $d$-dimensional Euclidean space. The algorithm starts with a clustering where each input point is in its own cluster. The algorithm then successively merges pairs of clusters. When two clusters are merged, a new cluster consisting of the union of the merged clusters is created. The *unmerged* clusters at a given time of the execution of the algorithm are the clusters that have not been merged so far. More formally, at the start the set of unmerged clusters is the set of all clusters. Then, whenever

two clusters are merged, the newly created cluster is inserted to the set of unmerged clusters while the two merged clusters are removed from the set. The algorithm merges clusters until all the points are in one cluster.

To be a $\gamma$-approximation to Average-Linkage, our algorithm must merge clusters according to the following rule: If the minimum average distance between a pair of unmerged clusters is $v$ then the algorithm is not allowed to merge two unmerged clusters with average distance larger than $\gamma \cdot v$.

Let $\varepsilon > 0$ and $\gamma \geqslant 1$ be parameters. We will show how to use a $\gamma$-approximate nearest neighbor data structure (on points) to get a $\gamma'$-approximate Average-Linkage algorithm where $\gamma' = (1+\varepsilon) \cdot \gamma$.

We make use of the following key ingredients.

- We design a sampling scheme that allows to choose at most poly $\log n$ points per cluster while preserving the average distance up to $(1+\varepsilon)$-factor with probability at least $1 - 1/n^5$.

- We design a data structure that given a set of clusters, allows to answer approximate nearest neighbor queries (on clusters) according to the average distance.

- Finally we provide a careful scheme for the merging steps that allows to bound the number of times the nearest neighbor queries for a given cluster have to be performed.

### 3.3   The Algorithm

We are now ready to describe our algorithm. Our algorithm starts with all input points in their own clusters and performs a nearest neighbor query for each of them. The algorithm maintains a partition of the input into clusters that we call the unmerged clusters, identical to average linkage. The algorithm proceeds in steps. Each step consists of merging several pairs of clusters. For each step we associate a value $v$, which we refer to as the *merge value of the step*, which is a power of $(1 + \varepsilon)$ and we will show the invariant that at the end of the step associated with value $v$, the unmerged clusters are at distance greater than $v/((1 + \varepsilon)^2 \gamma)$. Let $\mathcal{I}$ be the set of all merge values.

For each cluster $C$, we will maintain a sample of its points by applying the sampling procedure (see supplementary materials for more details). To avoid recomputing a sample too often, we set a variable $s(C)$ which corresponds to the size of the cluster the last time the sampling procedure was called.

**Lazy sampling.**   Every time two clusters $C_1, C_2$ are merged by the algorithm to create a new cluster, the following operations are performed:

1. If $|C_1 \cup C_2| \geqslant (1 + \varepsilon^2/(1+\gamma)) \max(s(C_1), s(C_2))$, then the sampling procedure is called on $C_1 \cup C_2$ and an approximate nearest cluster query is performed using the nearest cluster data structure (see supplementary materials). Then, $s(C_1 \cup C_2)$ is set to $|C_1 \cup C_2|$. The resolution parameter for sampling is the value of the current step divided by $n$. Namely, if the value of the current step is $v$, we set $\alpha_{C_1 \cup C_1} = v$ for the sampling procedure.

2. Otherwise, $s(C_1 \cup C_2)$ is set to $\max(s(C_1), s(C_2))$ and the algorithm uses the sample of $\text{argmax}_{C \in \{C_1, C_2\}} |C|$ as the sample for $C_1 \cup C_2$.

Once the above has been performed, a $\gamma$-approximate nearest cluster query is performed using the sample defined for the cluster resulting of the merge.

Thus, at each step, all the clusters have a $\gamma(1 + O(\varepsilon))$-approximate nearest neighbor among the clusters. We denote $\nu_t(C)$ the approximate nearest neighbor for cluster $C$ at the $t$th step. This approximate nearest neighbor is computed using our data structure (see supplementary materials). We let $\nu(C) = \nu_{t(C)}(C)$, where $t(C)$ is the step at which $C$ was created.

**Pseudocode for our algorithm**

1. Let $\mathcal{L}$ be the list of unmerged clusters, initially it contains all the points.

2. For each $v \in \mathcal{I}$:

    (a) ToMerge $\leftarrow \mathcal{L}$

    (b) While ToMerge is not empty:

i. Pick a cluster $C$ from ToMerge, and remove it from ToMerge.
ii. $NN(C) \leftarrow$ Approximate Nearest Neighbour Cluster of $C$.
iii. If $\text{avg}(C, NN(C)) \leqslant v$:
   A. Merge $C$ and $NN(C)$. Let $C'$ be the resulting cluster.
   B. Perform the Lazy Sampling procedure on $C'$ and insert it into the ANN data structure.
   C. Remove $NN(C)$ from ToMerge and add $C'$ to ToMerge;
   D. Remove $C, NN(C)$ from $\mathcal{L}$ and add $C'$ to $\mathcal{L}$

See supplementary materials for the proof of correctness.

## 4   Experiments

Our experiments focus on Ward's method and its approximation since it is a simpler algorithm in contrast with average-linkage. We implemented our algorithm using `C++11` on 2.5 GHz 8 core CPU with 7.5 GiB under the Linux operating system. Our algorithm takes a dynamic Nearest Neighbour data structure as a block box. In our implementation, we are using the popular `FLANN` library [30] and our own implementation of LSH for performing approximate nearest neighbor queries. We compare our algorithm to the sci-kit learn implementation of Ward's method [34] which is a Python library that also uses `C++` in the background.

Our algorithm has different parameters for controlling the approximation factor. These parameters have a significant effect on the performance and the precision of the algorithm. The main parameter that we have is $\epsilon$ which determines the number of data structures to be used (recall that we have one approximate nearest neighbor data structure for each $(1 + \varepsilon)^i$, for representing the potential cluster sizes) and the sequence of merge values. Moreover, we make use of `FLANN` library procedure for finding approximate nearest neighbors using KD-trees. This procedure takes two parameters the *number of trees* $t$ and the *number of leaves visited* $f$. The algorithm builds $t$ randomized KD-trees over the dataset. The number of leaves parameter controls how many leaves of the KD-trees are visited before stopping the search and returning a solution. These parameters control the speed and precision of the nearest neighbor search. For instance, increasing the number of leaves will lead to a high precision but at the expense of a higher running time. In addition, decreasing the number of KD-Tree increases the performance but it decreases the precision. For LSH, we use the algorithm of Datar et al. [18] which has mainly two parameters, $H$ the number of hash functions used and $r$ controlling the 'collision' rate (see details in [18]).

To study the effects of these parameters, we did different experiments that combine several parameters and we report and discuss the main results in Table 1. The main data that is used in these experiments are classic real-world datasets from the UCI repository and the sci-kit-learn library. Iris contains 150 points in 4 dimensions, Digits 1797 in 64 dimensions, Boston 506 points in 13 dimensions, Cancer 569 points in 3 dimensions, and Newsgroup 11314 points in 2241 dimensions.

To measure the speed-up achieved by our algorithm, we focus our attention on a set of parameters which gives classification error that is similar to Ward's on the real-world datasets, and then run our algorithm (with these parameters) on synthetic dataset of increasing sizes. These parameters are precisely $\epsilon = 8$, number of trees $T = 2$, the number of visited leaves $L = 10$. The datasets are generated using the blobs procedure of sci-kit learn. The datasets generated are $d$-dimensional for $d = \{10, 20\}$ and consists of a number of points ranging from 10 000 to 20 000. In both dimensions, we witness a significant speed-up over the sci-kit learn implementation of Ward's algorithm. Perhaps surprisingly, the speed-up is already significant for moderate size datasets. We observe that the running time is similar for LSH or FLANN.

**Acknowledgements.**   Ce projet a bénéficié d'une aide de l'État gérée par l'Agence Nationale de la Recherche au titre du Programme FOCAL portant la référence suivante : ANR-18-CE40-0004-01.

## Footnotes

[1] These lower bounds hold under the Strong Exponential Time Hypothesis of Impagliazzo and Paturi [23, 24] regarding the complexity of $k$-SAT.

[2] On the negative side, we know that a $(1 + \varepsilon)$ approximation requires quadratic time [36].

## References

[1] James Abello, Panos M Pardalos, and Mauricio GC Resende. *Handbook of massive data sets*, volume 4. Springer, 2013.

| | Iris | Cancer | Digits | Boston | Newsgroup |
|---|---|---|---|---|---|
| Ward's | 0.67 | 0.46 | 0.82 | 0.80 | 0.146 |
| Ward-FLANN ($\epsilon = 0.5$, $T = 16$, $L = 5$) | 0.62 | 0.53 | 0.79 | 0.80 | $< 0.05$ |
| Ward-FLANN ($\epsilon = 4$, $T = 16$, $L = 128$) | 0.76 | 0.47 | 0.56 | 0.78 | $< 0.05$ |
| Ward-FLANN ($\epsilon = 8$, $T = 2$, $L = 10$) | 0.75 | 0.51 | 0.47 | 0.80 | $< 0.05$ |
| Ward-LSH ($\epsilon = 10$, $r = 3$, $H = n^{1/10}$) | 0.69 | 0.58 | 0.58 | 0.82 | $< 0.05$ |
| Ward-LSH ($\epsilon = 10$, $r = 3$, $H = n^{1/2}$) | 0.72 | 0.48 | 0.73 | 0.83 | 0.104 |
| Ward-LSH ($\epsilon = 2$, $r = 3$, $H = n^{1/2}$) | 0.72 | 0.57 | 0.63 | 0.83 | 0.113 |

Table 1: We report the normalized mutual information score of the clustering output by the different algorithms compared to the ground-truth labels for each dataset. We note that 0.05 can obtained on Newsgroup through a random labelling of the vertices (up to $\pm 0.02$). Hence LSH seems a more robust approach for implementing approx-ward.

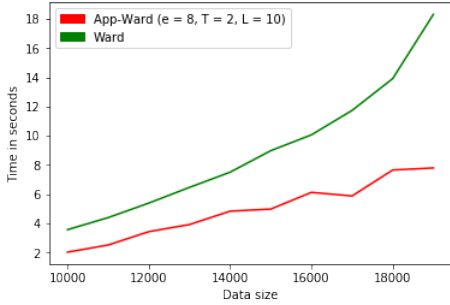

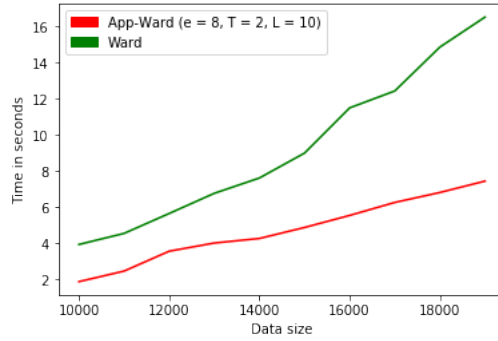

(a) Running time of our algorithm with parameters ($\varepsilon = 8$, T = 2, L = 10) (in red) and of Ward's method, on datasets of sizes ranging from 10 000 points to 20 000 points in $\mathbb{R}^{10}$. We observe that our algorithm is more than 2.5 faster on datasets of size 20 000.

(b) Running time of our algorithm with parameters ($\varepsilon = 8$, T = 2, L = 10) (in red) and of Ward's method, on datasets of sizes ranging from 10 000 points to 20 000 points in $\mathbb{R}^{20}$. We observe that our algorithm is more than 2.5 faster on datasets of size 20 000. Interestingly, it seems that the dimension has little influence on both our algorithm and Ward's method.

[2] Josh Alman and Ryan Williams. Probabilistic polynomials and hamming nearest neighbors. In *IEEE 56th Annual Symposium on Foundations of Computer Science, FOCS 2015, Berkeley, CA, USA, 17-20 October, 2015*, pages 136–150, 2015.

[3] Alexandr Andoni, Piotr Indyk, Huy L Nguyen, and Ilya Razenshteyn. Beyond locality-sensitive hashing. In *Proceedings of the twenty-fifth annual ACM-SIAM symposium on Discrete algorithms*, pages 1018–1028. Society for Industrial and Applied Mathematics, 2014.

[4] Alexandr Andoni and Ilya Razenshteyn. Optimal data-dependent hashing for approximate near neighbors. In *Proceedings of the forty-seventh annual ACM symposium on Theory of computing*, pages 793–801. ACM, 2015.

[5] Arturs Backurs, Piotr Indyk, and Ludwig Schmidt. On the fine-grained complexity of empirical risk minimization: Kernel methods and neural networks. In *Advances in Neural Information Processing Systems 30: Annual Conference on Neural Information Processing Systems 2017, 4-9 December 2017, Long Beach, CA, USA*, pages 4311–4321, 2017.

[6] Maria-Florina Balcan, Avrim Blum, and Santosh Vempala. A discriminative framework for clustering via similarity functions. In *Proceedings of the fortieth annual ACM symposium on Theory of computing*, pages 671–680. ACM, 2008.

[7] Allan Borodin, Rafail Ostrovsky, and Yuval Rabani. Subquadratic approximation algorithms for clustering problems in high dimensional spaces. In *Proceedings of the thirty-first annual ACM symposium on Theory of computing*, pages 435–444. ACM, 1999.

[8] Peter Breyne and Marc Zabeau. Genome-wide expression analysis of plant cell cycle modulated genes. *Current opinion in plant biology*, 4(2):136–142, 2001.

[9] Gunnar Carlsson and Facundo Mémoli. Characterization, stability and convergence of hierarchical clustering methods. *Journal of machine learning research*, 11(Apr):1425–1470, 2010.

[10] Moses Charikar and Vaggos Chatziafratis. Approximate hierarchical clustering via sparsest cut and spreading metrics. In *Proceedings of the Twenty-Eighth Annual ACM-SIAM Symposium on Discrete Algorithms*, pages 841–854. Society for Industrial and Applied Mathematics, 2017.

[11] Moses Charikar, Vaggos Chatziafratis, and Rad Niazadeh. Hierarchical clustering better than average-linkage. In *Proceedings of the Thirtieth Annual ACM-SIAM Symposium on Discrete Algorithms*, pages 2291–2304. SIAM, 2019.

[12] Moses Charikar, Vaggos Chatziafratis, Rad Niazadeh, and Grigory Yaroslavtsev. Hierarchical clustering for euclidean data. *arXiv preprint arXiv:1812.10582*, 2018.

[13] Ke Chen. On coresets for k-median and k-means clustering in metric and euclidean spaces and their applications. *SIAM Journal on Computing*, 39(3):923–947, 2009.

[14] Michael Cochez and Hao Mou. Twister tries: Approximate hierarchical agglomerative clustering for average distance in linear time. In *Proceedings of the 2015 ACM SIGMOD international conference on Management of data*, pages 505–517. ACM, 2015.

[15] Vincent Cohen-Addad, Varun Kanade, and Frederik Mallmann-Trenn. Hierarchical clustering beyond the worst-case. In *Advances in Neural Information Processing Systems*, pages 6201–6209, 2017.

[16] Vincent Cohen-Addad, Varun Kanade, Frederik Mallmann-Trenn, and Claire Mathieu. Hierarchical clustering: Objective functions and algorithms. In *Proceedings of the Twenty-Ninth Annual ACM-SIAM Symposium on Discrete Algorithms*, pages 378–397. SIAM, 2018.

[17] Sanjoy Dasgupta. A cost function for similarity-based hierarchical clustering. *arXiv preprint arXiv:1510.05043*, 2015.

[18] Mayur Datar, Nicole Immorlica, Piotr Indyk, and Vahab S Mirrokni. Locality-sensitive hashing scheme based on p-stable distributions. In *Proceedings of the twentieth annual symposium on Computational geometry*, pages 253–262. ACM, 2004.

[19] Ibai Diez, Paolo Bonifazi, Iñaki Escudero, Beatriz Mateos, Miguel A Muñoz, Sebastiano Stramaglia, and Jesus M Cortes. A novel brain partition highlights the modular skeleton shared by structure and function. *Scientific reports*, 5:10532, 2015.

[20] Pasi Franti, Olli Virmajoki, and Ville Hautamaki. Fast agglomerative clustering using a k-nearest neighbor graph. *IEEE transactions on pattern analysis and machine intelligence*, 28(11):1875–1881, 2006.

[21] Jerome Friedman, Trevor Hastie, and Robert Tibshirani. *The elements of statistical learning*, volume 1. Springer series in statistics New York, NY, USA:, 2001.

[22] Sariel Har-Peled, Piotr Indyk, and Rajeev Motwani. Approximate nearest neighbor: Towards removing the curse of dimensionality. *Theory of computing*, 8(1):321–350, 2012.

[23] Russell Impagliazzo and Ramamohan Paturi. On the complexity of k-sat. *Journal of Computer and System Sciences*, 62(2):367–375, 2001.

[24] Russell Impagliazzo, Ramamohan Paturi, and Francis Zane. Which problems have strongly exponential complexity? *Journal of Computer and System Sciences*, 63(4):512–530, 2001.

[25] Yongkweon Jeon, Jaeyoon Yoo, Jongsun Lee, and Sungroh Yoon. Nc-link: A new linkage method for efficient hierarchical clustering of large-scale data. *IEEE Access*, 5:5594–5608, 2017.

[26] Karthik C. S. and Pasin Manurangsi. On closest pair in euclidean metric: Monochromatic is as hard as bichromatic. In *10th Innovations in Theoretical Computer Science Conference, ITCS 2019, January 10-12, 2019, San Diego, California, USA*, pages 17:1–17:16, 2019.

[27] Meelis Kull and Jaak Vilo. Fast approximate hierarchical clustering using similarity heuristics. *BioData mining*, 1(1):9, 2008.

[28] Jure Leskovec, Anand Rajaraman, and Jeffrey David Ullman. *Mining of massive datasets*. Cambridge university press, 2014.

[29] Benjamin Moseley and Joshua Wang. Approximation bounds for hierarchical clustering: Average linkage, bisecting k-means, and local search. In *Advances in Neural Information Processing Systems*, pages 3094–3103, 2017.

[30] Marius Muja and David G. Lowe. Scalable nearest neighbor algorithms for high dimensional data. *Pattern Analysis and Machine Intelligence, IEEE Transactions on*, 36, 2014.

[31] Fionn Murtagh. A survey of recent advances in hierarchical clustering algorithms. *The Computer Journal*, 26(4):354–359, 1983.

[32] Fionn Murtagh. Comments on 'parallel algorithms for hierarchical clustering and cluster validity'. *IEEE Trans. Pattern Anal. Mach. Intell.*, 14(10):1056–1057, 1992.

[33] Dr Otair et al. Approximate k-nearest neighbour based spatial clustering using kd tree. *arXiv preprint arXiv:1303.1951*, 2013.

[34] F. Pedregosa, G. Varoquaux, A. Gramfort, V. Michel, B. Thirion, O. Grisel, M. Blondel, P. Prettenhofer, R. Weiss, V. Dubourg, J. Vanderplas, A. Passos, D. Cournapeau, M. Brucher, M. Perrot, and E. Duchesnay. Scikit-learn: Machine learning in Python. *Journal of Machine Learning Research*, 12:2825–2830, 2011.

[35] Aurko Roy and Sebastian Pokutta. Hierarchical clustering via spreading metrics. In *Advances in Neural Information Processing Systems*, pages 2316–2324, 2016.

[36] Aviad Rubinstein. Hardness of approximate nearest neighbor search. In *Proceedings of the 50th Annual ACM SIGACT Symposium on Theory of Computing*, pages 1260–1268. ACM, 2018.

[37] Hinrich Schütze, Christopher D Manning, and Prabhakar Raghavan. *Introduction to information retrieval*, volume 39. Cambridge University Press, 2008.

[38] Hinrich Schütze and Craig Silverstein. Projections for efficient document clustering. In *ACM SIGIR Forum*, volume 31, pages 74–81. ACM, 1997.

[39] Michael Steinbach, George Karypis, Vipin Kumar, et al. A comparison of document clustering techniques. In *KDD workshop on text mining*, volume 400, pages 525–526. Boston, 2000.

[40] Michele Tumminello, Fabrizio Lillo, and Rosario N Mantegna. Correlation, hierarchies, and networks in financial markets. *Journal of Economic Behavior & Organization*, 75(1):40–58, 2010.

[41] Pelin Yildirim and Derya Birant. K-linkage: A new agglomerative approach for hierarchical clustering. *Advances in Electrical and Computer Engineering*, 17(4):77–89, 2017.

