[Supplementary Material · main-11-16.pdf]

# A Missing Proofs and details

## A.1 Proofs for the approximate Ward's algorithm

*Proof of Lemma 2.1.* The running time follows almost immediately from the definition: there are $O(\varepsilon^{-1}\log n)$ data structure to query. The correctness results from the following argument. Consider the cluster $C^*$ that has been inserted to the data structure and that minimizes $\min_{C_0 \text{ inserted}} \Delta ESS(C, C_0)$. Let $j$ be the integer such that $(1+\varepsilon)^{j-1} \leqslant |C^*| \leqslant (1+\varepsilon)^j$. Consider the cluster $C_j$ returned by the query on $\mathcal{D}^j$. We have that $|C_j| \leqslant (1+\varepsilon)|C^*|$ and so by the correctness of the data structure $\Delta ESS(C_j, C) \leqslant \gamma(1+\varepsilon)\Delta ESS(C, C^*)$ and the lemma follows. $\qquad \square$

## A.2 Runtime analysis and correctness for the approximate Ward's algorithm

**Running Time** The outer loop of Algorithm 1 iterates $\beta$ times. The total number of clusters created by the algorithm is $O(n)$ where $n$ is the total number of input points. Thus, The inner for loop takes $O(n)$ times. By Lemma 2.1, the body of the inner loop will have at most the complexity of the nearest neighbour search $O(n^{f(\gamma)}\varepsilon^{-1}\log(n\Delta))$. Summing up all these complexities results in $O(n^{1+f(\gamma)}\varepsilon^{-1}\log(n\Delta))$.

**Proof of Correctness**

**Lemma A.1.** *Invariant 2.2 holds.*

*Proof.* We proceed by induction on the merge $\nu$. When the merge value is 1, the invariant trivially holds.

Now assume that the invariant holds up to some merge value $\nu$. We first show that there is no pair of clusters $C_i, C_j$ with $\Delta ESS(C_i, C_j) < \nu/\gamma$ at the end of the iteration corresponding to merge value $\nu$. Assume toward contradiction that this wasn't the case and consider the cluster of $C_i, C_j$ that was created the last, say $C_i$. Then, a nearest neighbor cluster query was made on $C_i$ and since $C_j$ was already in the data structure, Lemma 2.1 implies that the query returned a cluster of $C_\ell$ such that $\Delta(C_\ell, C_i) < \nu$. Hence $C_i$ was merged to $C_\ell$ and not an unmerged cluster at the end of the iteration.

$\qquad \square$

## A.3 Proofs for the approximate Average-Linkage algorithm

*Proof of Lemma 3.1.* Let $U = |A||C||B|$. We note that for each $a \in A, c \in C$, the triangle inequality implies that $d(a, c) \leqslant \min_{b \in B}(d(a, b) + d(b, c))$ and so $d(a, c) \leqslant \frac{1}{|B|}\sum_{b \in B}(d(a, b) + d(b, c))$.

$$
\begin{aligned}
\mathrm{avg}(A, C) &= \frac{1}{|A||C|}\sum_{a \in A}\sum_{c \in C} d(a, c) \\
&\leqslant \frac{1}{|A||C|}\sum_{a \in A}\sum_{c \in C}\frac{1}{|B|}\sum_{b \in B}(d(a, b) + d(b, c)) \\
&= \frac{1}{U}\sum_{a \in A}\sum_{c \in C}\sum_{b \in B}(d(a, b) + d(b, c)) \\
&= \frac{1}{U}\left(|C|\sum_{a \in A}\sum_{b \in B}d(a, b) + |A|\sum_{c \in C}\sum_{b \in B}d(b, c)\right) \\
&= \mathrm{avg}(A, B) + \mathrm{avg}(B, C)
\end{aligned}
$$

$\qquad \square$

### A.3.1 Approximating Cluster Distance by Sampling

Let $C_1, \ldots, C_k$ be a collection of clusters. Let $n^2\alpha_i$ be an upper bound on the average distance between points within $C_i$. Assume that the minimum average distance between any pair of clusters is at least $\alpha_i/n^2$ for all $i$. For each cluster $C_i$, we make a slight abuse of notation and let $\mathrm{avg}(C_i)$ denote the average distance between points in $C_i$ (i.e.: $\mathrm{avg}(C_i) = \mathrm{avg}(C_i, C_i)$). Let $c_i$ be a point

such that $\text{avg}(c_i, C_i) \leqslant \text{avg}(C_i)/\varepsilon$ and let $R_i$ denote the points of $C_i$ whose distance to $c_i$ is at most $\text{avg}(C_i)/\varepsilon^2$. In other words, $R_i = \{p \mid p \in C_i,\ \text{dist}(p, c_i) \leqslant \text{avg}(C_i)/\varepsilon^2\}$. Let $G_i = C_i - R_i$.

We consider the following sampling scheme. Among the points in $R_i$, pick $\eta\varepsilon^{-6}\log^3 n$ points uniformly at random. Let $\kappa_i = \text{avg}(G_i, R_i)$. By an immediate averaging argument we have that $|G_i| \leqslant \varepsilon|C_i|$.

We make use of the following lemma by Chen [13].

**Lemma A.2** ([13], Lemma 3.3)**.** *Let $V$ be a set of points in a metric space $(X, d)$, and let $\lambda', \xi > 0$ be given parameters. Let $\Delta$ be the diameter of $V$. Let $U$ be a sample of size $\xi^{-2}\ln(2/\lambda')$ points of $V$ picked independently and uniformly, where each point of $U$ is assigned weight $|V|/|U|$ such that $\sum_{u \in U} w(u) = |V|$. For a fixed point $p$, where $p$ is not necessarily a an element of $V$, we have that $|\sum_{v \in V}\text{dist}(v, p) - \sum_{u \in U} w(u)\text{dist}(u, p)| \leqslant \xi|V|\Delta$, with probability at least $1 - \lambda'$.*

From this, we deduce the following corollary.

**Corollary 1.** *Let $V$ be a set of points in a metric space $(X, d)$, and let $\lambda', \xi > 0$ be given parameters. Let $\Delta$ be the diameter of $V$. Let $U$ be a sample of size $\xi^{-2}\ln(2/\lambda')$ points of $V$ picked independently and uniformly. For a fixed point $p$, where $p$ is not necessarily a an element of $V$, we have that $|\text{avg}(V, p) - \text{avg}(U, p)| \leqslant \xi\Delta$, with probability at least $1 - \lambda'$.*

The proof of the following lemma is in the appendix.

**Lemma A.3.** *Given a set of point $C_i$ of size $m$, the sampling procedure can be performed in time $O(m/\varepsilon^5)$.*

For any two clusters $C_i, C_j$ let $S(C_i), S(C_j)$ denote the set of points sampled by the above procedure. Furthermore, we define $\widehat{\text{avg}}(C_i, C_j) = \text{avg}(S(C_i), S(C_j)) + \varepsilon\kappa_i + \varepsilon\kappa_j$. We then have the following crucial lemma, proved in the appendix.

**Lemma A.4.** *Consider a set of clusters $\{C_1, \ldots, C_\ell\}$ such that for any pair of clusters $C_i, C_j$, $\text{avg}(C_i), \text{avg}(C_j) \leqslant \eta\,\text{avg}(C_i, C_i)$ for some constant $\eta$.*

*Then, by taking a sampling of size $10\eta\varepsilon^{-6}\log^3 n$, we have $\widehat{\text{avg}}(C_i, C_j) = (1 \pm \varepsilon)\text{avg}(C_i, C_j)$ with probability at least $1 - 1/n^5$.*

### A.3.2 A Data Structure for Approximate Nearest Cluster

In this section, we introduce a data structure for finding approximate nearest clusters. The following theorem is proved in the appendix.

**Theorem A.5.** *Let $\gamma > 0$ be a parameter, $P$ a set . Let $\mathcal{D}$ be a data structure that for any set $P$ of $n$ points in $\mathbb{R}^d$ where $d = \Omega(\log n)$, supports the following operations:*

    *1. Insertion of a point in $P$ in time $O(n^{f(\gamma)})$, for some function $f$;*

    *2. Deletion of a point in $P$ in time $O(n^{f(\gamma)})$;*

    *3. Given a point $p \in P$, outputs a point inserted to the data structure at $L_1$-distance at most $\gamma$ times the distance from $p$ to the closest point inserted to the data structure, in time $O(n^{f(\gamma)})$.*

*Then, for any $\varepsilon > 0$, there exists a data structure for pairs $(S, w)$ where $S$ is a set of points in $\mathbb{R}^d$ and $w$ is a positive value, that supports the following operations:*

    *1. Insertion of a pair (set, value) in time $O(\eta\varepsilon^{-1}\log n \cdot n^{f(\gamma)})$;*

    *2. Deletion of a pair (set, value) in time $O(\eta\varepsilon^{-1}\log n \cdot n^{f(\gamma)})$;*

    *3. Given a set of points $C$ in $\mathbb{R}^d$ and a value $w$, outputs a pair $(C', w')$ inserted to the data structure that is such that that $\text{avg}(C, C') + w + w'$ is at most $\gamma(1 + \varepsilon)$ times $\min_{(C*, w*)\ \text{in the data structure}} \text{avg}(C, C^*) + w + w^*$ in time $O(\eta\varepsilon^{-1}\log n \cdot n^{f(\gamma)})$.*

*Proof of Lemma A.3.* We claim that we can simply use a constant factor approximation to the median problem to find $c_i$ – there is a vast literature of near-linear algorithms producing an $O(1)$-approximation to the median.

513 Consider the median of $P$, namely the point $p^* \in P$ that minimizes $\sum_{p \in P} \mathrm{dist}(p, p^*)$. We have that
514 $\frac{1}{|P|-1} \sum_{p \in P} \mathrm{dist}(p, p^*)$ is at most $\mathrm{avg}(P)$. Thus, consider any point $\hat{p}$ that is an $O(1)$-approximation
515 to the median of $P$. We have that $\frac{1}{|P|-1} \sum_{p \in P} \mathrm{dist}(p, \hat{p}) = O(\mathrm{avg}(P))$.

516 Then, the remaining step of the sampling procedure is to evaluate the distance from each point to $\hat{p}$
517 to define $R_i$ and $G_i$. This can be done in linear time. Finally, the sampling of points in $R_i$ can also
518 be done in linear time. $\qquad\square$

519 *Proof of Lemma A.4.* We have, by Lemma 3.1,

$$\mathrm{avg}(C_i, C_j) = \frac{|R_i|}{|C_i|}\mathrm{avg}(R_i, C_j) + \frac{|G_i|}{|C_i|}\mathrm{avg}(G_i, C_j)$$

$$\leqslant \mathrm{avg}(R_i, C_j) + \frac{|G_i|}{|C_i|}\mathrm{avg}(R_i, G_i)$$

$$\leqslant \mathrm{avg}(R_i, C_j) + \varepsilon \cdot \mathrm{avg}(R_i, G_i)$$

$$\leqslant \mathrm{avg}(R_i, C_j) + \varepsilon \cdot \mathrm{avg}(C_i)$$

520 Similarly, we have

$$\mathrm{avg}(R_i, C_j) = \frac{|R_j|}{|C_j|}\mathrm{avg}(R_i, R_j) + \frac{|G_j|}{|C_j|}\mathrm{avg}(G_j, R_i)$$

$$\leqslant \mathrm{avg}(R_i, R_j) + \frac{|G_j|}{|C_j|}\mathrm{avg}(R_j, G_j)$$

$$\leqslant \mathrm{avg}(R_i, R_j) + \varepsilon \cdot \mathrm{avg}(R_j, G_j)$$

$$\leqslant \mathrm{avg}(R_i, R_j) + \varepsilon \cdot \mathrm{avg}(C_j)$$

Combining yields

$$\mathrm{avg}(C_i, C_j) \leqslant \mathrm{avg}(R_i, R_j) + \varepsilon \cdot \mathrm{avg}(C_j) + \varepsilon \cdot \mathrm{avg}(C_i).$$

521 Therefore, by applying Corollary 1 to $S(C_i), S(C_j)$, we have that $\mathrm{avg}(S(C_i), S(C_j)) = (1 \pm$
522 $\varepsilon)\mathrm{avg}(R_i, R_j)$ and so $\mathrm{avg}(C_i, C_j) \leqslant (1 + \varepsilon)\widehat{\mathrm{avg}}(C_i, C_j)$ since the diameter of the points in $R_i$
523 and $R_j$ is at most $\mathrm{avg}(C_i)/\varepsilon$ and $\mathrm{avg}(C_j)/\varepsilon$ respectively.

524 We now aim at proving that $\mathrm{avg}(C_i, C_j) \geqslant (1 - O(\varepsilon\eta))\widehat{\mathrm{avg}}(C_i, C_j)$. Recall that by assumption,
525 we have that $\mathrm{avg}(C_i), \mathrm{avg}(C_j) \leqslant \eta \cdot \mathrm{avg}(C_i, C_j)$. Thus, again combining with Corollary 1, we
526 have that $\widehat{\mathrm{avg}}(C_i, C_j) \leqslant (1 + \varepsilon)\mathrm{avg}(R_i, R_j) + 2\varepsilon\eta \cdot \mathrm{avg}(C_i, C_j)$. Moreover, as discussed above,
527 we have that $\mathrm{avg}(C_i, C_j) \geqslant (1 - O(\varepsilon))\mathrm{avg}(R_i, R_j)$ and so, rescalling $\varepsilon$, we have $\widehat{\mathrm{avg}}(C_i, C_j) \leqslant$
528 $(1 + \varepsilon)\mathrm{avg}(C_i, C_j)$, as claimed.

529 $\qquad\square$

530 *Proof of Theorem A.5.* We start with some preprocessing steps and notations. We consider an iso-
531 metric embedding of all the input points into $L_1$ with distortion at most $(1 + \varepsilon)$, for some sufficiently
532 small $\varepsilon > 0$. In the remaining, we thus work with the $L_1$ norm.

533 For each point $p$, for each integer $i$, let $p^i = \underbrace{p_1 \cdot p_1 \dots p_1}_{i}$ Namely, the coordinates of $p^i$ are ob-
534 tained by concatenating the coordinates of $p$ $i$ times. Given a set of $j$ points $S = \{p_1, p_2, \dots, p_j\}$
535 and a value $w_S$, we let $q^i(S)$ be the point in a $(i \cdot j \cdot d + 2)$-dimensional space with coor-
536 dinates $p_1^i, p_2^i, \dots p_j^i, 0, i \cdot j \cdot w_S$. Namely, $q^i(S)$ is obtained by concatenating $p^i$ of all the $j$
537 points $p \in S$, adding an extra coordinate of value $0$ and adding a final coordinate with value
538 $i \cdot j \cdot w_S$. We also let $d^i(S)$ be the point in a $(j \cdot i \cdot d + 2)$-dimensional space with coordinates
539 $\underbrace{p_1, p_2, \dots p_j, p_1, p_2, \dots p_j, \dots p_1, p_2, \dots p_j}_{i \cdot j}, i \cdot j \cdot w_S, 0$. Namely obtained by concatenating the co-
540 ordinates of the point $p_1, p_2, \dots p_j$, $i$ times, adding an extra coordinate of value $i \cdot j \cdot w_S$ and adding a

final coordinate with value 0. We have the following claim, whose proof follows immediately from the definition.

**Claim 1.** *Given two sets $A$ and $B$, of size $i$ and $j$ respectively, and two values $w_A, w_B$, we have that*

$$\frac{1}{i \cdot j}||q^j(A) - d^i(B)||_1 = w_A + w_B + \frac{1}{i \cdot j}\sum_{a \in A}\sum_{b \in B}||a - b||_1.$$

We now describe our data structure using an approximate nearest-neighbor data structure $\mathcal{D}$ for the $L_1$ distance between points. We make use of an approximate nearest neighbor data structure $\mathcal{D}^{i,j,k}$, for each integers $i, j \in \{1, 2, \ldots, \eta\}$.

Let $C$ be a cluster. The insertion is as follows. Let $i = |C|$. The algorithm inserts the point $d^j(C)$ in the data structure $\mathcal{D}^{i,j}$, for all $j \in \{1, 2, \ldots, \eta\}$. Deletion of $C$ consists of removing $d^j(C)$ from the $\mathcal{D}^{i,j}$ it has been inserted into. The time complexities for insertion and deletion follow immediately.

The approximate nearest neighbor query for cluster $C$ is performed as follows. For all $j \in \{1, 2, \ldots, \eta\}$, the algorithm creates the point $q^j(C)$, and makes a nearest neighbor query in the data structure $\mathcal{D}^{i,j}$. Let $p^j$ be the point returned by the query $q^j(C)$ on data structure $\mathcal{D}^{i,j}$ and $\nu^j$ be the cluster corresponding to $p^j$. Claim 1 implies that $\frac{1}{|C| \cdot |\nu^j|}||q^j(C) - d^i(\nu^j)||_1 = (1 \pm \varepsilon)(w_C + w_{\nu^j} + \mathrm{avg}(C, \nu^j))$.

Then, let $j^* = \mathrm{argmin}_j \, \mathrm{avg}(C, \nu^j)$. We now argue that $\mathrm{avg}(C, \nu^{j^*}) + w_C + w_{\nu^{j*}} \leqslant \gamma(1 + \varepsilon)\min_{C' \neq C}(\mathrm{avg}(C, C') + w_C + w_{C'})$.

Let $\hat{C} = \mathrm{argmin}_{C' \neq C}(\mathrm{avg}(C, C') + w_C + w_{C'})$ and $\hat{j} = |\hat{C}|$. Consider the data structure $\mathcal{D}^{i,\hat{j}}$. By its correctness, $\mathcal{D}^{i,\hat{j}}$ returned a point $p^{\hat{j}}$ such that $||q^{\hat{j}}(C) - p^{\hat{j}}||_1 \leqslant \gamma(||q^{\hat{j}}(C) - d^i(\hat{C})||_1)$. Thus, applying Claim 1 yields that $\mathrm{avg}_w(C, \nu^{\hat{j}}) + w_C + w_{\nu^{\hat{j}}} \leqslant \gamma(1 + \varepsilon)(\mathrm{avg}(C, \hat{C}) + w_C + w_{\hat{C}})$. By the choice of $j^*$, we thus have that $\mathrm{avg}(C, \nu^{j^*}) \leqslant \gamma\mathrm{avg}(C, \hat{C})$, as claimed. $\qquad\square$

**Invariant.** The correctness of the algorithm is captured by the following invariant. The proof, as well as the running time analysis, are deferred to the appendix.

**Lemma A.6** (Invariant for correctness). *The following holds with probability at least $1 - 1/n^3$. Consider the $t$th step of the algorithm, let $v$ be the merge value at the $t$th step.*

1. *At the end of the step, no cluster at (inner) average distance greater than $v(1 + \varepsilon)$ has been merged by the algorithm so far. For any unmerged clusters $C_i, C_j$, we have that $\widehat{\mathrm{avg}}(C_i, C_j) = (1 + O(\varepsilon))\mathrm{avg}(C_i, C_j)$.*

2. *For any unmerged cluster $C$ at the end of the step, $\nu_t(C)$ is an unmerged $(1 + O(\varepsilon))\gamma$-approximate nearest cluster of $C$.*

3. *Finally, at the end of a step of value $v$, there is no pair of clusters at average distance less than $v/((1 + \varepsilon)^2\gamma)$.*

*Proof of Lemma A.6.* We prove it by induction on the number of steps of the algorithm. This is clearly true at first.

We start with (1). For simplicity, assume that first that the algorithm does not do lazy sampling and runs the sampling procedure after each merge. Then, (1) follows from the definition of the algorithm and the inductive hypothesis on the correctness of the sampling procedure (Lemma A.4). More formally, the definition of the algorithm ensures that no pair of clusters $C_i, C_j$ such that $\widehat{\mathrm{avg}}(C_i, C_j) > v(1 + \varepsilon)$ are merged Moreover, by the inductive hypothesis, we have that for any cluster $C$, $\mathrm{avg}(C) \leqslant v(1 + \varepsilon)$.

Thus, we can apply Lemma A.4 with $\eta = (1 + \varepsilon)$ and we deduce that for any pair of clusters $C_i, C_j$, $\widehat{\mathrm{avg}}(C_i, C_j) = (1 \pm \varepsilon)\mathrm{avg}(C_i, C_j)$ with probability at least $1 - 1/n^5$. Taking a union bound over all $n$ steps and $n$ merges of the algorithm and all $O(n^2)$ pairs of clusters in total concludes the proof of (1) in the case of non-lazy sampling.

To finish the proof of (1), we need to show that lazy sampling does not degrade the quality of the outcome of the sampling by too much. Hence, consider an unmerged cluster resulting from the merge possibly at a previous step of two clusters $C_1, C_2$. If $|C_1 \cup C_2| \geqslant (1 + \varepsilon^2/(1 + \gamma)) \max(s(C_1), s(C_2))$, then the sampling procedure is applied and the average distance between the samples of $C_1 \cup C_2$ and any other cluster $C_3$ is within a $(1 + \varepsilon)$ factor from the average distance between $C_1 \cup C_2$ and $C_3$ with probability at least $1 - 1/n^4$ and the above analysis applies. Now, if $|C_1 \cup C_2| < (1 + \varepsilon^2/(1 + \gamma)) \max(s(C_1), s(C_2))$, then assume w.l.o.g. that $|C_1| \geqslant |C_2|$. Hence, we have that by Lemma 3.1 that for any other unmerged cluster $C_3$ $\mathrm{avg}(C_2, C_3) \leqslant \mathrm{avg}(C_2, C_1) + \mathrm{avg}(C_1, C_3)$. Now, by the inductive hypothesis, we have that $\mathrm{avg}(C_2, C_1) \leqslant \gamma \mathrm{avg}(C_1, C_3)$ and so $\mathrm{avg}(C_2, C_3) \leqslant (1 + \gamma)\mathrm{avg}(C_1, C_3)$. It follows that $\mathrm{avg}(C_1 \cup C_2, C_3) \leqslant (1 + \varepsilon)\mathrm{avg}(C_1, C_3)$. Finally, by the induction hypothesis, we have that the sample of $C_1$ preserves the distance from $C_1$ to $C_3$ with probability at least $1 - 1/n^4$ up to a $(1 + \varepsilon)$ factor. Thus, we indeed have that the average distance between the samples of any pair of unmerged clusters is within a factor $(1 + \varepsilon)$ of the average distance of the pair.

We then turn to (3), thus consider the end of a step. Observe that if there are two clusters $C_1, C_2$ that are at pairwise distance less than $v/((1 + \varepsilon)^2 \gamma)$ then by the inductive hypothesis, the sampling procedure guarantees the two samples for $C_1, C_2$ are at average distance at most $v/\gamma$. Therefore, a $\gamma$-approximate nearest cluster query returns a cluster at distance less than $v$. Thus, consider the cluster, say $C_2$, that is inserted into the data structure last. When $C_2$ is processed, a nearest neighbor query is performed and so, since the cluster $C_1$ has been inserted first in the data structure, $C_2$ should have had an approximate nearest neighbor at distance less than $v$ and so should have been merged.

We now move to prove (2): We finish by considering unmerged clusters at step $t$. We show that for any unmerged cluster $C$, the nearest cluster is at average distance at least $\frac{1}{\gamma}\mathrm{avg}(C, \nu(C))$ and at most $(1 + 1/n)\mathrm{avg}(C, \nu(C))$. This will conclude the proof of the invariant.

Let $i$ be the step at which $C$ is created. Let $C^*$ be the nearest cluster to $C$ at the $t$th step. By Theorem A.5, Lemma A.4 and the inductive hypothesis of the $\gamma$-approximate nearest neighbor procedure we have that $\mathrm{avg}(\nu(C), C) \leqslant \gamma \mathrm{avg}(C^*, C)$. Since the unmerged clusters at step $t > i$ are the union of the clusters of $C^i$, we have that $\mathrm{avg}(C, C_0) \geqslant \mathrm{avg}(C, C^*)$ for any cluster $C_0$ of $C^{t_0}$. It follows that for any $i' \geqslant i$, the cluster of $C^{i'}$ that is the nearest to $C$ is at distance at least $\gamma^{-1} \cdot \mathrm{avg}(C, \nu(C))$.

We now show an upper bound on the distance to the cluster $C'$ containing $\nu(C)$. This follows from applying Lemma 3.1 as follows. Consider the sequence of merges that involve $\nu(C)$. Let $\nu(C) \subset \nu(C)_1 \subset \ldots \subset \nu(C)_k$ denote the clusters that contain $\nu(C)$ and that are successively merged after step $i$ and until time $t$. By Lemma 3.1, we have that $\mathrm{avg}(C, \nu(C)_1) \leqslant \mathrm{avg}(C, \nu(C)) + \mathrm{avg}(\nu(C), \nu(C)_1) \leqslant \mathrm{avg}(C, \nu(C)) + \mathrm{avg}(C, \nu(C))/n^2$ since $C$ is not active. Similarly, by the inductive hypothesis (1), $\mathrm{avg}(C, \nu(C)_2) \leqslant \mathrm{avg}(C, \nu(C)_1) + \mathrm{avg}(\nu(C)_1, \nu(C)_2)$. Here again, $C$ is not active and so $\mathrm{avg}(\nu(C)_1, \nu(C)_2) \leqslant \mathrm{avg}(C, \nu(C))/n^2$. Since the overall number of merges is at most $n$, we conclude that $\mathrm{avg}(C, \nu(C)_k) \leqslant +\mathrm{avg}(C, \nu(C)) + \mathrm{avg}(C, \nu(C))/n$ as claimed.

Therefore, the invariant also holds and so the inductive hypothesis is satisfied. $\square$

## A.4 Running Time Analysis for the approximate Average-Linkage algorithm

We need to bound the number of times an approximate nearest cluster query is performed, the total time incurred by the sampling procedure, the running time of a step, and the number of steps. This is the purpose of the following section.

### A.4.1 Sampling Time

Lemma A.7 bounds the total running time incurred by the sampling procedure.

**Lemma A.7.** *The total running time caused by the sampling procedure over the entire execution of the algorithm is at most $O(n^{1+\rho}\varepsilon^{-2}\gamma \log n)$.*

*Proof.* The lemma follows from Lemma A.3 and due to the fact that the procedure is only called on clusters resulting from the merge of two clusters $C_1, C_2$ such that $|C_1 \cup C_2| \geqslant (1 + \varepsilon^2/(1 + \gamma)) \max(s(C_1), s(C_2))$. Thus, the number of clusters in which an input point can contribute to the running time of the sampling procedure is $O(\varepsilon^{-2}\gamma \log n)$. $\square$

633 **Running Time of a Step**  At a given step associated with a certain merge value $v$, the goal is to
634 merge all clusters whose nearest neighbor is at distance at most $v$ so that at the end of the step,
635 the distance from each cluster to its approximate nearest neighbor is greater than $v$. Let $n_v$ be the
636 number of active clusters at the beginning of the step.

637 **Lemma A.8.** *The total number of nearest neighbor queries made by the algorithm during a step*
638 *with merge value $v$ is $O(n_v)$.*

639 *Proof.* Observe that the total number of merges is at most $O(n_v)$. Moreover the total number of
640 nearest neighbor queries is bounded by the total number of merges plus the number of active clusters
641 and so at most $O(n_v)$.

642 $\square$

643 A cluster can remain active throughout the entire algorithm. Hence, the number of active step is a
644 priori only bounded by $O(\varepsilon^{-1} \log \Delta n)$ which gives the claimed complexity.

645 A slightly more involved algorithm allows to remove the dependency in $\log \Delta$ at the price of a
646 slightly worse approximation guarantee: we were only able to show a $\gamma^2$-approximation instead of
647 a $\gamma$-approximation in this case. We defer this to the full version of the paper.