[Reviews · NeurIPS 2019]

Reviewer 1



This paper proposes a new approach to approximating hierarchical agglomerative clustering (HAC) by requiring that at each round, only a gamma-best merge be performed (gamma being the multiplicative approximation factor to the closest pair). Two algorithms are introduced to approximate HAC - one for Ward and one for Average linkage. In both cases, the algorithms rely on using approximate nearest neighbor ANN as a black box. In addition, a bucketing datastructure is used in Wards algorithm and a subsampling procedure in used for Average linkage to guarantee the subquadratic runtime. This is a new contribution to the theoretical literature on HAC, a provable subquadratic algorithm for (an approximation to) HAC cases other than single linkage. The paper demonstrates the performance of Ward’s approximation on real and synthetic datasets. The paper is well organized. There were parts where clarity could be improved. Specifically, here are some suggestions / typos: -59 fist -> first -95: Meaning of epsilon not clear. E.g. How does epsilon trade off with accuracy. Role of epsilon vs gamma becomes clear later, but it might help to explain this here (or simplify to a single parameter) -223: follows -> can be computed -Parts of section 3.2 seems to overlap with section 2 - e.g. description of general HAC procedure -Concept of merge value explained twice in sections 2 and 3 - can be combined -272/273 - resolution parameter alpha not defined / introduced, this part is unclear. -Explanation of the average linkage approximation algorithm was hard to follow. Pseudocode would be helpful Regarding the experimental section, the range of parameter tuning (3 combinations of epsilon, T, and L) does not seem comprehensive. Furthermore choosing a single combination as ‘best’ for evaluating the runtime does not seem fully justified. Finally, the speedup of 2.5X (on n=20K points and d=10) does not seem that impressive.

Reviewer 2



- Strength: This paper is well written and the ideas are clearly presented, which makes the paper easy to follow. The algorithm seems easy to implement and is likely to have some value in practice. - Weakness: My main concern is about the problem itself. The proposed method uses ANN to accelerate NN query, and claims achieving some "constant approximation". The "approximation" here means that in each step the two merged clusters differ only by constant times of the minimum "dissimilarity". However, with such relaxation, the output hierarchical tree can be completely different from the exact solution. It is not clear from the paper how to compare the quality of these two hierarchical trees. Obviously, if the two trees are not close, the acceleration would not make much sense. Although the proposed algorithm runs faster, it may not achieve the desired goal. The paper does give some empirical evidence (Table 1) to suggest that the approximate solution is of "good quality". But the experiment is conducted only on a few small datasets and thus not very convincing. There are a few papers discussing how to define a suitable objective function for the HC problem (e.g. [1] and [2]). I would like to see some discussions and comparisons with those approaches. - [1] Sanjoy Dasgupta. A cost function for similarity-based hierarchical clustering. STOC'16 - [2] Vincent Cohen-Addad et al. Hierarchical Clustering: Objective Functions and Algorithms. SODA'17

Reviewer 3



The authors present sub-quadratic algorithms for hierarchical clustering. In particular they present an efficient version of the average linkage algorithm and of the Ward algorithm. The main idea behind the algorithm is to slightly relax the decision principle used by these algorithms in every step and then to show that there exists efficient algorithms for the relaxed version of the problem. Interestingly, the authors also argue(via a reductions to closest pair) that it is unlikely to obtain fast algorithms without relaxing the decision principles. The main idea behind the algorithm is to use LSH to compute approximate nearest neighbors and to then use LSH to obtain fast algorithm. Unfortunately, LSH cannot be applied directly so the authors carefully design algorithms to obtain good theoretical guarantees. I found the results for average linkage particularly interesting. Minor comment: - ln. 189 and 191, I found confusing to use i for both indices - I find unmerged cluster a confusing name because the set may contained merged clusters - ln273, where is defined \alpha_{C_1\cup C_2} - ln.483, 488. It is odd to point to the appendix in the appendix

[Author Response · NeurIPS 2019]

We thank all the reviewers for their time and for their thoughtful comments. We agree with all that was said and will do our best to address it in the final version. In particular, we will address the valuable suggestions on the presentation (including adding pseudocode) and we agree that it would be interesting to run our algorithms on larger datasets, which will likely increase the gap in running time between the classic $n^2$ implementation and our algorithm, as the theory predicts and our first experiments show. We will run the experiments on KDDCUP99 and NEWSGROUP and report them in the final version. The reason we did not prioritize this initially is that, to us, the main value of the experiments is a proof of concept that our notion of approximation leads to good clusters (to be discussed more below) rather than to highlight the speedup (which directly follows from the theoretical gap in complexity and the efficiency of LSH in practice).

The focus of this response will be to discuss the following important concern raised by reviewer #2. We will discuss a few points that were mentioned too briefly in the paper (or not at all), but that will be included in the full version.

**On a theoretical justification for our notions of approximation (a concern raised by reviewer #2)**

The approximate Average-Linkage notion that we define ($\gamma$-AL) guarantees that at every step, the merged pair is $\gamma$-close to the best one. But can we prove any guarantees on the quality of the final tree? Will it be "close" to the output of (exact) AL? (Same is true for Ward's, but let us focus on AL in this response.)

One approach that we have considered that also seems to be what the reviewer has in mind is to look at certain objective functions that measure the quality of a hierarchical clustering tree, and compare the guarantees of AL and our $\gamma$-AL w.r.t. these objective functions. Such functions were proposed by [2], and [4] (and by [3] for similarity graphs). It is likely that one can prove that $\gamma$-AL is guaranteed to give a solution that is no worse than an $O(\gamma)$ factor from the guarantees of (exact) AL w.r.t. to these objective functions. However, such a theorem may not have much value because (as shown by Charikar et al. [1]) the guarantees of AL are no better than those of a random recursive partitioning of the dataset. Therefore, such a theorem will only prove that $\gamma$-AL is not-much-worse than random, which dramatically understates the quality of $\gamma$-AL. In fact, in our experiments with a standard classification task, $\gamma$-AL is very close to AL and is *much* better than random (random has a $1/k$ success rate, which is $0.3$ or less, while ours achieves $0.5 - 0.8$).

Another approach would be to prove theorems pertaining to an objective function for HC that offers the guarantee that given two trees, if their costs are close then the structures of their HCs are similar. Unfortunately, we are not aware of any such objective functions (this is also the case for flat clusterings such as k-median, k-means, etc.). In particular, with the functions of [2, 4] the trees output by AL and by a random recursive partitioning have the same cost, while their structure may be very different.

Besides the empirical evidence which, despite being on a small dataset, we find to be a promising proof of concept that our approximate notions make sense, let us mention two more reasons (that we will add to the paper) for the value of our algorithms:

First, our algorithm is essentially a reduction to Approximate Nearest Neighbor (ANN) queries, and ANN queries (using LSH for example) perform very well in practice. In fact, on real world inputs, the algorithm often identifies the *exact* nearest neighbor and then performs the same merge as in AL.

Second, we can provide a theoretical analysis of the following form in support of $\gamma$-AL. It is known that if the input data is an ultrametric, then AL (and also Single-Linkage or Complete-Linkage) does recover the underlying ultrametric tree (see e.g.: Cohen-Addad et al.) . Now, assume that the ultrametric is 'clear' in the sense that if $d(a, b) > d(a, c)$ then $d(a, b) > \gamma d(a, c)$ for some constant $\gamma$. In this case, our algorithm will provably recover the ultrametric in $n^{1+O(1/\gamma)}$ time, whereas AL would need $\Omega(n^2)$ time. Notably, in this setting, obtaining an $O(1)$-approximation w.r.t. the objective functions of [2, 4] does not mean that the solution is close to the ultrametric tree.

*We wish to sincerely thank the reviewers and the PC again for their time and help in improving the quality of this work.*

# References

[1] M. Charikar, V. Chatziafratis, and R. Niazadeh. Hierarchical clustering better than average-linkage. In *Proceedings of the Thirtieth Annual ACM-SIAM Symposium on Discrete Algorithms*, pages 2291–2304. SIAM, 2019.

[2] V. Cohen-Addad, V. Kanade, F. Mallmann-Trenn, and C. Mathieu. Hierarchical clustering: Objective functions and algorithms. In *Proceedings of the Twenty-Ninth Annual ACM-SIAM Symposium on Discrete Algorithms*, pages 378–397. SIAM, 2018.

[3] S. Dasgupta. A cost function for similarity-based hierarchical clustering. *arXiv preprint arXiv:1510.05043*, 2015.

[4] B. Moseley and J. Wang. Approximation bounds for hierarchical clustering: Average linkage, bisecting k-means, and local search. In *Advances in Neural Information Processing Systems*, pages 3094–3103, 2017.


[Meta-Review · NeurIPS 2019]

Thanks for your submission to NeurIPS. This paper was very much a borderline paper, with two accept scores and one reject score. One of the concerns raised by the negative reviewer was that, while the algorithm can achieve an approximation to the best merge at each step, it is unclear how the final clustering results would compare to the standard algorithm. The authors addressed this in their rebuttal, which helped. Also, there were some issues raised about experiments as well as various minor suggestions (typos etc.). In general it seems that the concerns are mostly minor, and on the whole this paper seems to make an interesting and worthwhile contribution, so I am recommending that the paper is accepted.